# Ultrasonic Localization of Transformer Patrol Robot Based on Wavelet Transform and Narrowband Beamforming

**DOI:** 10.3390/s25185723

**Published:** 2025-09-13

**Authors:** Hongxin Ji, Zijian Tang, Jiaqi Li, Chao Zheng, Xinghua Liu, Liqing Liu

**Affiliations:** 1School of Electrical Engineering, China University of Mining and Technology, Xuzhou 221116, China; ts23230145p31@cumt.edu.cn (Z.T.); ts23230113p31@cumt.edu.cn (J.L.); ts22230196p31@cumt.edu.cn (C.Z.); 2College of Mechanical and Electronic Engineering, Shandong Agricultural University, Tai’an 271018, China; lxh9357@163.com; 3State Grid Tianjin Electric Power Research Institute, Tianjin 300180, China; liulq328@126.com

**Keywords:** patrol robot, robot localization, signal denoising, wavelet decomposition, ultrasonic array localization

## Abstract

The large size and metal-enclosed casings of oil-immersed power transformers present significant challenges for patrol robots attempting to accurately locate their position within the transformer. Therefore, this paper proposes a three-dimensional spatial localization method for transformer patrol robots using a nine-element ultrasonic array. This method is based on wavelet decomposition and weighted filter beamforming (WD-WFB) algorithms. To address the issue of strong noise interference in the field, the ultrasonic localization signals are adaptively decomposed into wavelet coefficients at different frequencies and scales. An improved semi-soft thresholding function is applied to the decomposed wavelet coefficients to reduce noise and reconstruct the localization signals, resulting in localization signals with low distortion and a high signal-to-noise ratio(SNR). To overcome the limitations of traditional beamforming algorithms regarding interference resistance and signal resolution, this paper presents an improved WFB algorithm. By obtaining the energy distribution of the scanning area and determining the position of the maximum energy point, the spatial position of the transformer patrol robot can be determined. The test results show that the proposed improved semi-soft threshold function demonstrates superior denoising performance compared to traditional threshold functions. When compared to the soft threshold function, it achieves improvements of 15.32% in SNR and 15.57% in normalized correlation coefficient (NCC), along with a 48.91% reduction in root mean square error (RMSE). Compared with the hard threshold function, the improvement is even more significant: the SNR is improved by 60.55%, the NCC is improved by 24.90%, and the RMSE is reduced by 58.77%. The denoising effect was significantly improved compared to the traditional threshold function. In a 1200 mm × 1000 mm × 1000 mm transformer test box, the improved WFB algorithm in this paper was used to perform multiple localizations of the transformer patrol robot at different positions after denoising the field signals using the semi-soft threshold function. The maximum relative localization error was 3.47%, and the absolute error was within 2.6 cm, meeting engineering application requirements.

## 1. Introduction

The large power transformer is one of the most critical pieces of hub equipment in a power system. Taking various preventive measures in advance to evaluate the status of large transformers can help avoid major power accidents. These measures are crucial for improving the safe and stable operation of power systems. However, due to the metal sealing of large transformers, which causes poor internal visibility, it is difficult to effectively and intuitively determine the internal insulation status of large transformers through the analysis of dissolved gases in oil alone (e.g., the three-ratio method and improved three-ratio method) without dismantling large transformers [1,2,3]. Internal insulation conditions mainly include damage, discoloration, deformation, displacement, and dendritic discharge marks of the winding enclosure; damage, deformation, and cracking of the insulating pressing plate; foreign matter, corrosion, and discharge on the inner surface of the oil tank; discharge traces of internal magnetic shielding device; discharge carbon traces on the insulation support; and a degree of impurity contamination in the transformer oil.

With the progress of artificial intelligence and micro robot technology, using a small patrol robot to detect transformer internals can achieve faster and more intuitive detection of transformer faults [4,5,6,7,8]. Accurately locating the three-dimensional position of a patrol robot is key to ensuring that it can avoid obstacles and reach the target point for defect detection. Because the metal shell completely seals a transformer, which is filled with transformer oil, there is no light. Therefore, it is difficult to use conventional methods such as laser radar, visual cameras, and GPS to locate micro patrol robots [9,10,11,12,13]. However, the ultrasonic location method has the advantages of no lighting requirements, a wide detection range in transformer oil, and high location accuracy. Therefore, ultrasonic localization is a practical and feasible method for patrol robots.

Current ultrasonic sensor array positioning techniques primarily include time delay estimation, high-resolution spectral analysis, and controllable beamforming. Time delay estimation involves two stages: first, the time differences of arrival (TDOA) between the sound source and sensors are determined using algorithms such as generalized cross-correlation (GCC) or least mean square (LMS) adaptive filtering [14,15,16,17,18]; second, the sound source location is estimated through a spatial search based on the TDOA and the known sensor geometry. However, this method is highly susceptible to performance degradation caused by reverberation and ambient noise, which can lead to substantial localization errors, especially under conditions of strong reverberation or low SNR.

High-resolution spectral analysis techniques process signals from ultrasonic arrays by arranging them into a specific geometric pattern. Then, they determine the source’s angle and distance by analyzing the spatial spectrum. Typical methods include the autoregressive (AR) model and the maximum entropy (ME) approach [19]. These techniques assume that the signals are narrowband and originate from a far-field source with plane wavefronts and that the sensors have consistent characteristics. They involve complex calculations, such as matrix operations and covariance matrix inversion. Furthermore, their effectiveness may be impaired by reflected signals and interference.

Beamforming-based positioning uses the weighted sum of ultrasonic array signals to create directional beams. The location of the sound source corresponds to the spatial coordinate that maximizes the output power of the beam, which is determined by scanning potential source positions [20,21]. Although it possesses inherent interference rejection capabilities and is widely used for two-dimensional source localization, conventional beamforming algorithms exhibit low spatial resolving power and inadequate suppression of substantial noise interference. Due to the three-dimensional nature of patrol robot positioning and the confined, high-noise environment within transformers, traditional beamforming is not easily applicable to achieving accurate three-dimensional localization.

To overcome this difficulty, this paper proposes a three-dimensional localization method for a transformer patrol robot with a nine-element ultrasonic array based on the WD-WFB algorithm. The patrol robot sends narrowband ultrasonic signals that are received by the positioning array and denoised using a wavelet algorithm. Then, the signal is imaged using the improved WFB algorithm, which ultimately achieves the intuitive and accurate localization of the transformer patrol robot.

## 2. Positioning Principle of Transformer Patrol Robot

### 2.1. Ultrasonic Propagation and Attenuation in Transformer Oil

In a fluid medium, the ultrasonic wave is an elastic longitudinal wave. Its propagation speed depends on the medium’s density and elastic constant. The sound speed in the transformer oil can be expressed as follows:(1)$$c=\sqrt{\frac{\partial p}{\partial \rho }}=\sqrt{\frac{1}{\rho K_{a}}}$$
where p is pressure, c is the sound speed in the transformer oil, and ρ is the density of the transformer oil. $K_{a}$ is the adiabatic compression coefficient of the transformer oil. Because ρ and $K_{a}$ are functions of temperature and static pressure, temperature has the most significant effect. Therefore, considering the internal ambient temperature T and static pressure (H) of the transformer, the ultrasonic velocity can be corrected by the following empirical equation:(2)$$c=1449.2+4.6T-0.055T^{2}+0.016H$$

As the sound wave propagates through transformer oil, it gradually weakens with the increase in propagation distance. Main attenuation includes geometric wavefront expansion, boundary loss, absorption, and scattering. The absorption coefficient can be expressed as follows:(3)$$\alpha =\frac{16\pi^{2}\mu_{s}f^{2}}{3\rho c^{3}}$$
where α is the sound intensity absorption coefficient, $\mu_{s}$ is the shear viscosity, and f is the frequency. For a fluid medium and small amplitude conditions, the acoustic wave equation of an acoustic wave can be expressed as(4)$$\left\{\begin{array}{l} \frac{\partial u}{\partial t}+\frac{1}{\rho }gradp=0\\ \frac{\partial p}{\partial t}+\rho c^{2}divu=0 \end{array}\right.$$
where u is the velocity vector of the fluid particle, t  is time, *gradp* represents the pressure gradient, which is related to fluid acceleration and described by Euler’s equation, and *divu* represents the velocity divergence, which is related to the compressibility of the medium and is described by the continuity equation. Assuming that the density of the transformer oil is uniform, the three-dimensional spatial wave equation can be expressed as follows:(5)$$\left(\frac{\partial^{2}}{\partial x^{2}}+\frac{\partial^{2}}{\partial y^{2}}+\frac{\partial^{2}}{\partial z^{2}}\right)p-\frac{1}{c^{2}}\frac{\partial^{2}p}{\partial t^{2}}=0$$
where x, y, and z are mutually perpendicular coordinate axes in three-dimensional space. During the propagation of a sound wave in a medium, if the distance between the center of the sound source and the receiving point is much greater than the size of the sound source, the sound source can be approximated as a point sound source (monopole). Therefore, the sound pressure of the sound field can be expressed as follows:(6)$$p\left(r,t\right)=\frac{\rho c\frac{\omega }{c}{r_{0}}^{2}u_{a}}{r\sqrt{1+(\frac{\omega }{c}r_{0}{)}^{2}}}e^{j\left(\omega t-\frac{\omega }{c}r+\theta \right)}$$
where p(r, t) is the sound pressure at time t from the center r of the spherical source, *ρ* is the static density of the medium, c is the velocity at which the sound wave propagates in the medium, $r_{0}$ is the radius of the sound source, $u_{a}$ is the velocity at which the sound source vibrates, ω is the angular frequency, and θ is the initial phase angle of sound source vibration. Equation (6) shows that the distance between the signal source and the ultrasonic array sensor affects the signal’s phase and amplitude.

### 2.2. Narrowband Signal Model

In general, a narrowband signal is defined as one for which the time it takes the incident wave to pass through the array is much less than the inverse of the signal bandwidth. It has characteristics such as limited duration, limited energy, and limited frequency range:(7)$$\tau \ll \frac{1}{W_{B}}\;{\int_{-\infty }^{\infty }\left|s\left(t\right)\right|}^{2}dt<\infty$$
where τ is the time delay, $W_{B}$ is the signal’s bandwidth, and s(t) is the signal in the time domain. The narrowband signal model uses a simplified signal delay model to model the array data. In this model, the signal received by the sensor can be expressed by the following equation:(8)$$s\left(t\right)=u\left(t\right)e^{j\left[\omega t+\varphi \left(t\right)\right]}$$
where u(t) represents the signal amplitude, φ(t) represents the signal phase, and ω represents the angular frequency.

After a delay, the received signal can be represented as follows:(9)$$s\left(t-\tau \right)=u\left(t-\tau \right)e^{j\left[\omega \left(t-\tau \right)+\varphi \left(t-\tau \right)\right]}$$

For narrowband signals, ω  can be considered a constant. Then, the changes in u(t) and φ(t) can be ignored:(10)$$u\left(t-\tau \right)\approx u\left(t\right)\;\varphi \left(t-\tau \right)\approx \varphi \left(t\right)$$

Substitute (10) into (9) to get (11).(11)$$s\left(t-\tau \right)=u\left(t\right)e^{j\left[\omega t+\varphi \left(t\right)\right]}e^{-j\omega \tau }=s\left(t\right)e^{-j\omega \tau }$$

Equation (11) shows that a time delay for narrowband signals in the ultrasonic positioning array can be converted into a phase shift.

### 2.3. Positioning Principle of Array Beamforming

According to the geometric relationship, the time difference between the signal received by sensor *i* in the array and the signal received by reference sensor 1 can be calculated using sensor 1 as the reference array element:(12)$$\tau_{i}=\frac{d_{i}-d_{1}}{c}$$
where $\tau_{i}$ is the difference in ultrasonic arrival time between the *i*-th sensor and the reference sensor, $d_{1}$ is the distance from the sound source to sensor 1, and $d_{i}$ is the distance from the sound source to the *i*-th sensor.

Therefore, the signals received by the entire sensor array can be expressed as follows:(13)$$X\left(t\right)=\left[\begin{array}{c} s_{1}\left(t\right)\\ s_{2}\left(t\right)\\ \cdots \\ s_{m}\left(t\right) \end{array}\right]=\left[\begin{array}{c} \alpha_{1}e^{-j\omega \tau_{1}}\\ \alpha_{2}e^{-j\omega \tau_{2}}\\ \cdots \\ \alpha_{m}e^{-j\omega \tau_{m}} \end{array}\right]\left[s\left(t\right)\right]+\left[\begin{array}{c} n_{1}\left(t\right)\\ n_{2}\left(t\right)\\ \cdots \\ n_{m}\left(t\right) \end{array}\right]$$
where $X\left(t\right)$ is the received signal vector, $s_{m}\left(t\right)$ is the signal received by the *m*-th sensor, $\alpha_{m}$ is the attenuation coefficient of the m-th sensor, and $\tau_{m}$ is the propagation time delay from the sound source to the m-th sensor. Additionally, $s\left(t\right)$ is the signal received by the reference ultrasonic sensor, and $n_{m}\left(t\right)$ is the additive noise received by the m-th sensor at time t. This can be expressed in the matrix form as follows:(14)$$X\left(t\right)=a_{\theta }s\left(t\right)+N\left(t\right)$$
where $a_{\theta }$ denotes the directional orientation vector of the array, which is calculated based on the ultrasound-receiving sensing array layout, and $N\left(t\right)$ is the noise vector.

In the localization array, weighting coefficients are used to adjust the amplitude and time delay of each channel signal. The adjusted signals from each channel are then summed to obtain the output of the time delay-summing beamforming, as shown in Figure 1.

Figure 1 shows that the output signal y(t) of the time delay-summing beamforming algorithm can be represented as follows:(15)$$y\left(t\right)=W^{H}X\left(t\right)$$
where W is the weight vector, and $W=[k_{1}w_{1}(\tau_{1}),k_{2}w_{2}(\tau_{2}),k_{3}w_{3}(\tau_{3}),\ldots ,k_{m}w_{m}(\tau_{m}){]}^{T}$.

The power of the output signal of the time delay-summing beamforming algorithm can be expressed as follows:(16)$$G=E\{y(t{)}^{2}\}=W^{H}RW$$
where G is the output power, and R is the covariance matrix calculated using the localization array output X(t). In other words, $R=E\{X(t)X^{H}(t)\}$.

According to Equation (16), selecting a suitable weight vector to compensate for the time delay of the array signal makes it possible for the array to have the highest output power at the incident position of the sound source, while the power output at other positions is smaller. This allows one to determine the position of the sound source.

However, strong noise in the field can degrade positioning accuracy. To suppress the effects of reverberation and noise as much as possible, a WFB localization method is used.

The output of the improved WFB algorithm can then be expressed as follows:(17)$$Q=\int_{-\infty }^{+\infty }\frac{S\left(\omega \right)Y^{*}\left(\omega \right)}{\sqrt{S\left(\omega \right)S^{*}\left(\omega \right)}}e^{j\omega \tau }d\omega =\int_{-\infty }^{+\infty }S\left(\omega \right)Y^{*}\left(\omega \right)\phi \left(\omega \right)e^{j\omega \tau }d\omega$$
where Q represents the coherent output of the WFB algorithm, reflecting the signal correlation in different spatial directions. The greater the amplitude of Q, the stronger the signal correlation in that direction, indicating that the sound source may come from that direction. S(ω) is the Fourier transform of the reference sensor signal s(t), Y(ω) is the Fourier transform of the output signal y(t) of the time delay-summing beamforming algorithm, ω is the angular frequency, τ is the time delay, and ϕ(ω) is the weighting filter function.

Since noise and reverberation reduce the correlation between S(ω) and Y(ω), thereby reducing the peak amplitude of Q and possibly causing a deviation in the peak position, weighted filtering can be used to mitigate the effects of noise and reverberation, increase the peak amplitude of Q, and improve the accuracy of sound source localization.

### 2.4. Wavelet Packet Decomposition and Filtering

The wavelet transform of any signal x(t) is defined as the inner product of the signal and the wavelet basis function:(18)$$W_{x}\left(a,b\right)=<x\left(t\right),\psi_{ab}\left(t\right)\geq \int_{-\infty }^{+\infty }x\left(t\right)\psi_{ab}^{*}\left(t\right)\;dt$$(19)$$\psi_{ab}\left(t\right)=\frac{1}{\sqrt{a}}\psi \left(\frac{t-b}{a}\right)\;\;\;a>0$$
where $W_{x}(a,b)$ is the wavelet coefficient, and $\psi_{ab}(t)$ represents the wavelet basis function. Additionally, a is the scaling factor, and b is the translation parameter, and its value can be positive or negative. The symbol * represents complex conjugation.

The wavelet transform obtains the frequency characteristics of the ultrasonic signal by scaling the width of the wavelet basis function, and it obtains the time information of the signal by shifting the wavelet basis function. To compress data and reduce computation, the wavelet transform can be performed on discrete scales and translation values. When the scale and translation are discretized, the wavelet basis function $\psi_{ab}(t)$ can be rewritten as $\psi_{jk}(t)$:(20)$$\psi_{jk}\left(t\right)=a_{0}^{-\frac{j}{2}}\psi \left(a_{0}^{-j}t-k\right)\;\;\;\;j=0,1,2,\cdots ;k\in Z$$
where $a_{0}$ is the discretization scale factor, k is the discretization translation parameter, and j is the frequency domain. Generally, $a_{0}=2$, with the corresponding wavelet transform being(21)$$W_{x}\left(j,k\right)=2^{-\frac{j}{2}}\int x\left(t\right)\;\psi_{jk}^{*}\left(2^{-j}t-k\right)\;dt$$

The wavelet-based denoising process is mainly divided into three parts: decomposition, denoising, and reconstruction of the ultrasonic positioning signal. The schematic diagram of the localization signal denoising process is shown in Figure 2.

The essence of wavelet threshold denoising is to suppress useless parts of the signal while enhancing useful parts. The wavelet packet threshold denoising process is as follows:The decomposition process involves selecting a wavelet to perform n-layer wavelet decomposition on a signal;The threshold process involves threshold denoising for the decomposed coefficients of each layer;In the reconstruction process, the denoised wavelet coefficients are reconstructed to obtain the denoised signal.

The proper selection of threshold functions is essential for wavelet packet threshold denoising. Hard and soft thresholding functions are widely used.

The hard threshold function is expressed as(22)$$W_{hard}\left(j,k\right)=\left\{\begin{array}{l} W_{x}(j,k),\;\;\;\left|W_{x}\left(j,k\right)\right|\geq \lambda \\ 0,\;\;\;\;\;\;\;\;\left|W_{x}\left(j,k\right)\right|<\lambda \end{array}\right.$$
where $W_{hard}\left(j,k\right)$ is the wavelet coefficient value after hard thresholding. The wavelet coefficient remains unchanged when the absolute value of the wavelet coefficient $W_{x}(j,k)$ is greater than the given threshold λ. On the contrary, the wavelet coefficient is zero. The hard threshold better preserves the local features of the signal.

The soft threshold function is expressed as follows:(23)$$W_{soft}\left(j,k\right)=\left\{\begin{array}{l} \mathrm{sgn}(W_{x}(j,k))(\left|W_{x}(j,k)\right|-\lambda ),\;\left|W_{x}(j,k)\right|\geq \lambda \\ 0,\;\;\;\;\left|W_{x}\left(j,k\right)\right|<\lambda \end{array}\right.$$
where *sgn*( ) is a symbolic function, and $W_{soft}\left(j,k\right)$ is the wavelet coefficient value after soft thresholding. When the absolute value of the wavelet coefficient is greater than the given threshold, the coefficient is subtracted from the threshold. On the contrary, the wavelet coefficient is zero. The signal processed by the soft threshold function is smoother.

The hard threshold function exhibits discontinuities at ±*λ*, which induce oscillations in the reconstructed signal near these points, compromising its smoothness relative to the original signal. While the soft threshold function yields continuous wavelet coefficients, it introduces undesirable amplitude compression, which degrades the approximation accuracy between the reconstructed and true signals. To address these limitations, this paper proposes a novel semi-soft threshold function that integrates the advantages of both approaches:(24)$$W_{new}\left(j,k\right)=\left\{\begin{array}{l} W_{x}(j,k),\;\;\;\left|W_{x}\left(j,k\right)\right|\geq \lambda_{2}\\ \mathrm{sgn}\left[W_{x}\left(j,k\right)\right]\lambda_{2}{\left[\frac{\left|W_{x}\left(j,k\right)\right|-\lambda_{1}}{\lambda_{2}-\lambda_{1}}\right]}^{h},\;\lambda_{1}<\left|W_{x}\left(j,k\right)\right|<\lambda_{2}\\ 0,\;\;\;\;\;\;\;\;\;\left|W_{x}\left(j,k\right)\right|\leq \lambda_{1} \end{array}\right.$$
where $W_{new}\left(j,k\right)$ is the wavelet coefficient value after semi-soft thresholding, and $\lambda_{1}$ and $\lambda_{2}$ are the lower and upper thresholds of the threshold function, respectively, with $\lambda_{2}=2\lambda_{1}$. The threshold function characteristic curve is shown in Figure 3.

From a mathematical perspective, $\underset{\left|W_{x}(j,k)\right|\to {\lambda_{1}}^{+}}{\lim}W_{new}(j,k)=0$, $\underset{\left|W_{x}(j,k)\right|\to {\lambda_{2}}^{-}}{\lim}W_{new}(j,k)=W_{x}(j,k)$. In other words, the improved semi-soft threshold function is continuous at the segment points, which overcomes the disadvantage of the hard threshold function being discontinuous at the segment points.

Define the shrinkage rate as follows:(25)$$\eta =\frac{W_{new}\left(j,k\right)}{\left|W_{x}\left(j,k\right)\right|}$$

The soft and semi-soft threshold shrinkage rates are as follows:(26)$$\eta_{soft}=1-\frac{1}{\left|W_{x}\left(j,k\right)\right|}$$(27)$$\eta_{new}=\frac{\lambda_{2}}{\left|W_{x}\left(j,k\right)\right|}{\cdot \left[\frac{\left|W_{x}\left(j,k\right)\right|-\lambda_{1}}{\lambda_{2}-\lambda_{1}}\right]}^{h}$$

Equation (26) shows that the soft threshold shrinkage rate is a fixed linear relationship with deviation, making it unable to adapt to signal characteristics. Equation (27) shows that when $\left|W_{x}(j,k)\right|\to {\lambda_{2}}^{-}$, $\eta_{new}\;\to \;1$ and approaches no contraction. When $\left|W_{x}(j,k)\right|\to {\lambda_{1}}^{+}$, $\eta_{new}\;\to \;0$ and approaches complete suppression. Therefore, the improved semi-soft threshold function overcomes the linear bias issue present in the soft threshold function.

Therefore, the semi-soft threshold function eliminates wavelet coefficients below a lower threshold while preserving those above an upper threshold. This approach simultaneously addresses the discontinuity issue inherent in hard thresholding and the amplitude compression problem associated with soft thresholding.

## 3. Simulation Study on Beam Imaging Localization of Patrol Robots

According to the patrol robot positioning and imaging principle described above, this paper focuses on how the improved WFB algorithm, the shape of the sensor array, the distance between the sound source and the array, and the output order of the beamforming affect the imaging effect of the patrol robot. The localization process is shown in Figure 4.

### 3.1. Effect of the Improved WFB Algorithm

Set the sound source position of the transformer patrol robot as (400 mm, 300 mm, and 600 mm). The ultrasonic signal emitted by the patrol robot consists of three 180 kHz sinusoidal signals. The sampling frequency is 2.5 MSa/s, and the number of sampling points is 2000. Due to the complex background noise inside the transformer, Gaussian white noise with an SNR of 0 dB is added to the signal in order to simulate the effect of noise on the signal, as shown in Figure 5.

Wavelet decomposition is performed on the noisy signal to obtain wavelet coefficients. Then, the hard, soft, and semi-soft thresholding functions are used to denoise each wavelet coefficient. These denoised coefficients are then reconstructed to obtain the denoised signal, as shown in Figure 6.

In order to further demonstrate the effect of the improved threshold function, the SNR, RMSE, and NCC are used for evaluation. They are defined as follows:(28)$$SNR=10\;lg\frac{\sum_{t=1}^{N}{x\left(t\right)}^{2}}{\sum_{t=1}^{N}{\left(x\left(t\right)-y\left(t\right)\right)}^{2}}$$(29)$$RMSE=\sqrt{\frac{1}{N}\sum_{t=1}^{N}{\left(x\left(t\right)-y\left(t\right)\right)}^{2}}$$(30)$$NCC=\frac{\sum_{t=1}^{N}x\left(t\right)y\left(t\right)}{\sqrt{\sum_{t=1}^{N}{x\left(t\right)}^{2}\sum_{t=1}^{N}{y\left(t\right)}^{2}}}$$
where $x\left(t\right)$ represents the original signal, $y\left(t\right)$ represents the signal after noise removal, and N represents the length of the signal. The SNR, RMSE, and NCC for different denoising threshold functions are shown in Table 1.

Higher SNR values indicate reduced noise contamination and improved signal quality. Lower RMSE measurements correspond to minimized signal distortion, while NCC values approaching 1 signify superior denoising performance. Table 1 demonstrates that the semi-soft threshold function achieves optimal results: the highest SNR, the lowest RMSE, and the highest NCC values nearest to 1. These metrics collectively confirm the enhanced denoising capability of the semi-soft threshold function.

In order to analyze the localization effect of the improved beamforming algorithm, a circular ultrasonic array is used as an example. The sensor at the center of the ultrasonic positioning array is the coordinate origin, and its coordinates are set to (0,0,0). Figure 7 shows the coordinates of the other sensors, where the coordinate system parameter is *b* = 110 mm. The sound source position of the transformer patrol robot is set to (400 mm, 300 mm, and 600 mm), and the scanning area is set to a three-dimensional space of 1200 mm × 1000 mm × 1000 mm. The scanning step for the X-axis, Y-axis, and Z-axis is 2 mm.

The traditional beamforming algorithm and the improved WFB algorithm are used to simulate the localization performance of the sound source, respectively. The localization results are shown in Figure 8 and Figure 9.

Figure 8 and Figure 9 show slices of the 3D stereo spatial localization results at Z = 600 mm. Figure 8 shows the localization results using the traditional beamforming algorithm. It can be seen that there are multiple red, high-energy regions in the detection region, and the background noise is relatively large. In contrast, the detection area obtained by the improved WFB algorithm has only one red, high-energy area, and the background noise is relatively small. The coordinates of the point with the maximum energy value are (400 mm, 300 mm). This is consistent with the real position of the sound source, as shown in Figure 9.

Since the energy values in the red, high-energy regions are all very close to each other, maximum energy values may appear in different high-energy regions due to noise interference. Therefore, the larger the number of red, high-energy regions, the larger the localization error. Compared to the traditional beamforming algorithm, the WFB algorithm has only one high-energy region, and the background noise is smaller. Thus, the WFB algorithm has a better localization effect.

### 3.2. Effect of Sensor Array Shape on Positioning

Key parameters such as the array guidance vector and viewing area depend on the shape of the sensor array. This leads to the critical influence of array shape on positioning. In this paper, three typical array shapes are selected: circular ultrasonic array, L-shaped ultrasonic array, and square ultrasonic array. The number of sensors in each array is 5. The shape and geometric dimensions of the array are shown in Figure 10, Figure 11 and Figure 12.

Set the sound source position of the transformer patrol robot to (300 mm, 200 mm, 600 mm), and the scanning area to a three-dimensional space of 1200 mm × 1000 mm × 1000 mm. The scanning step for the X-axis, Y-axis, and Z-axis is 2 mm.

The localization performance of the three typical ultrasonic localization arrays in Figure 10, Figure 11 and Figure 12 is simulated using the improved WFB algorithm, and the localization results are shown in Figure 13, Figure 14 and Figure 15.

Figure 13, Figure 14 and Figure 15 show the slices of 3D stereo positioning results at Z = 600 mm. A high-energy area appears in the detection area. The coordinates of the maximum energy point are (400 mm, 300 mm), that is, the location of the sound source. Compared with the positioning results of the circular and L-shaped ultrasonic arrays, the area of high energy in the positioning results of the square ultrasonic array is smaller, and the positioning effect is best. Therefore, this paper uses a square array to locate the patrol robot of a transformer in three-dimensional space.

### 3.3. Effect of the Number of Sensors in the Ultrasonic Array

In order to analyze the influence of the number of sensors in the ultrasonic array on the localization effect of the transformer patrol robot, this paper selected four-element square arrays, five-element square arrays, and nine-element square arrays, respectively, as shown in Figure 16, Figure 17 and Figure 18.

The localization results for different array shapes are shown in Figure 19, Figure 20 and Figure 21.

Figure 19 shows a slice of the 3D spatial localization results of the four-element square array at Z = 600 mm. A cluster of high-energy regions appears in the detection region. Figure 20 shows a slice of the 3D spatial localization results of the five-element square array at Z = 600 mm. It can be seen that the number of high-energy regions is fewer. Figure 21 shows the three-dimensional spatial localization results of the nine-element square array at Z = 600 mm. It can be seen that there is only one high-energy region in the detection area. The energy value of this high-energy region is much larger than that of the surrounding area. The coordinates of the point with the largest energy value are (400 mm, 300 mm).

A comparison of Figure 19, Figure 20 and Figure 21 shows that the nine-element square array has the smallest area of high-energy regions and the best imaging and localization. As the number of sensors in the localization array decreases, the number of pseudo-high-energy regions appearing in the detection area becomes larger, the energy values become larger and closer, and the localization performance gradually decreases under the influence of noise.

### 3.4. Effect of Sound Source Position on Positioning

To analyze the influence of the distance between the sound source and the ultrasonic array on the imaging positioning of the transformer patrol robot, taking the nine-element square arrays as an example, to simulate the positioning performance of the sound source, three different positions of sound sources from far to near were set as (560 mm, 360 mm, 600 mm), (280 mm, 180 mm, 300 mm), and (140 mm, 90 mm, 150 mm), respectively, as shown in Figure 22, Figure 23 and Figure 24.

Figure 22, Figure 23 and Figure 24 show the slices of the 3D stereo positioning result. A high-energy area appears in the detection area. To sum up, when the sound source is close to the sensor, the area of the high-energy area in the detection area is small, and the imaging positioning effect is the best. With the increase in distance, the area of the high-energy area gradually becomes larger, which means that the positioning performance degrades.

### 3.5. Comparative Analysis of Different Localization Methods

To validate the effectiveness of the method described in this paper, the localization results of different localization methods were compared. The nine-element rectangular sensor array remained unchanged (the sensor arrangement parameters are shown in Figure 18, and the sound source position was set to (400 mm, 300 mm, 600 mm). The localization performances of the WD-WFB, MVDR, and GCC-PHAT algorithms were verified when SNR = 0 dB.

Figure 21 shows the localization results of the WD-WFB algorithm at an SNR of 0 dB: at Z = 600 mm, the coordinates of the maximum energy point are (400 mm, 300 mm). Thus, the localization results of the WD-WFB algorithm are consistent with the true position of the sound source (400 mm, 300 mm, 600 mm), with little interference in the surrounding area.

Similarly, scanning the spatial spectrum using the MVDR algorithm revealed that a maximum energy region existed at Z = 600 mm. A slice was taken at Z = 600 mm, and the results are shown in Figure 25.

Figure 25 shows that at Z = 600 mm, the coordinates of the maximum energy point are (400 mm, 300 mm). In other words, the localization result of the MVDR algorithm is (400 mm, 300 mm, 600 mm), which is the true position of the sound source. However, a comparison of the WD-WFB and MVDR algorithms reveals that while the MVDR algorithm can determine the true position of the sound source, its weak interference resistance leads to a large number of false energy points around the maximum energy point, making it difficult to distinguish them visually and increasing the likelihood of misjudgment. In contrast, the WD-WFB algorithm has fewer false energy points around the maximum energy point and is easier to distinguish visually.

When using the GCC-PHAT algorithm for localization, it is necessary to calculate the time delays between signal arrivals at different sensors. Since this study focuses on three-dimensional spatial localization, three variables (X, Y, and Z) must be solved for, requiring at least three sets of time delays and at least four sensors.

In this study, four sensors were selected for locating the sound source signal: U2 (110 mm, 110 mm, 0), U4 (110 mm, −110 mm, 0), U6 (−110 mm, −110 mm, 0), and U8 (−110 mm, 110 mm, 0). Using the U2 sensor as a reference, the theoretical time delays between the sound source signal propagating to different sensors are Δt_24_ = 61.72 μs, Δt_26_ = 134.44 μs, and Δt_26_ = 80.78 μs.

The GCC-PHAT algorithm is used to calculate time delays. Taking Δt_24_ as an example, the calculation equation is(31)$$R_{24}\left(\tau \right)=\frac{1}{2\pi }\int_{-\infty }^{+\infty }G_{24}\left(\omega \right)\Psi \left(\omega \right)e^{j\omega \tau }d\omega$$
where $G_{24}(\omega )$ is the cross-power spectral density function of the two-channel time-domain signals $x_{2}(t)$ and $x_{4}(t)$, mathematically expressed as $G_{24}\left(\omega \right)=X_{2}\left(\omega \right)X_{4}^{*}\left(\omega \right)$. Ψ(ω) is the GCC weighting function, and when using PHAT for weighting, $\Psi (\omega )\;=\frac{1}{\left|G_{24}(\omega )\right|}$.

Through calculation, Δt_24_ = 58.4 μs. Similarly, we can obtain Δt_26_ = 131.2 μs and Δt_28_ = 78.8 μs. Substituting the obtained time delay information into the localization equation system, the final localization results are (371.4 mm, 268.8 mm, 552.3 mm). Compared to the actual sound source location, the localization error is −28.6 mm in the X-axis direction, −31.2 mm in the Y-axis direction, and −47.7 mm in the Z-axis direction.

Therefore, compared to the WD-WFB algorithm and MVDR algorithm described in this paper, the GCC-PHAT algorithm exhibits a more noticeable error in localization accuracy.

## 4. Experimental Verification

### 4.1. Three-Dimensional Space Positioning Test Platform for Transformer Patrol Robots

In order to verify the practicability of the positioning method proposed in this paper, an experimental test platform was built, as shown in Figure 26. The test platform mainly includes a transformer patrol robot, an ultrasonic positioning array, data acquisition, and a control platform.

The tank used for the test platform is 1200 mm long, 1000 mm wide, and 1000 mm high. The transformer patrol robot and ultrasonic array are all in the tank, and the coordinates of the acoustic source emission sensor are (230 mm, 50 mm, 730 mm).

The size of the transformer patrol robot has been rigorously designed to ensure that it can be successfully adapted to the requirements of the environment inside the transformer.

In order to ensure the accurate localization of the patrol robot in the whole domain of the transformer, it is required that the ultrasonic signals emitted by the transformer patrol robot at different locations can be received by the localization array. This requires that the ultrasonic signals emitted by the sound source have wide directionality, while the localization array receives sensors with wide directionality. To this end, this paper designs a hemispherical piezoelectric ceramic ultrasound probe, which has a diameter of 12.7 mm, an operating frequency of 160~200 kHz, a resonance frequency of 180 kHz, a sensor vertical directionality of −90~90°, and a sensor horizontal directionality of 0~360°. The ultrasonic sensor array consists of nine omnidirectional waterproof piezoelectric ceramic ultrasonic probes; the probe distribution and coordinate positions are shown in Figure 27, in which the origin of the coordinate system is set at the center sensor with coordinates of (0,0,0), and the sensor parameter is b = 110 mm; the nine sensors are uniformly distributed.

The internal environment of the transformer is complex. In order to prevent collision, the maximum cruising speed of the designed transformer patrol robot is 0.06 m/s. Its relatively slow movement speed and localization frequency of over 3 Hz meet the engineering needs well. The localization algorithm in this paper runs the program in less time, meeting the real-time localization needs of the transformer patrol robot.

### 4.2. Experiment and Data Acquisition Process

The ultrasonic sensor on the top of the patrol robot sends out a group of ultrasonic driving signals (three consecutive sinusoidal signals) with a frequency of 180 kHz every 500 ms. At this time, the ultrasonic array receives the ultrasonic signal, and the data acquisition device synchronously collects it. The parameters of the data acquisition device are as follows: the sampling rate of each channel is 2.5 MSa/s, the acquisition depth is 4096 points, and the trigger mode is the rising edge trigger.

The acquisition device is shown in Figure 28, and the acquired ultrasonic waveform signal is shown in Figure 29.

The nine time-domain waveforms shown in the figure are ultrasonic waveform signals collected by sensors U1–U9. Mechanical equipment vibration, electromagnetic noise, and environmental white noise make the ultrasonic waveform signal contain system noise. In addition, the amplitude of the sine signal changes in the process of propagation, making it difficult to accurately obtain the position of the transformer patrol robot. Therefore, this paper decomposes the ultrasonic waveform signal into three layers of wavelet packets. Taking the signal of U1 as an example, the decomposed wavelet coefficients are shown in Figure 30.

Figure 30 shows that the valuable signal components at the a31 decomposition node are large. Therefore, this paper first uses the general threshold method to estimate each wavelet coefficient’s threshold value, and then uses the semi-soft threshold function to denoise each wavelet coefficient. Finally, the denoised wavelet coefficients are reconstructed, and the reconstructed ultrasonic waveform signal is shown in Figure 31.

After wavelet denoising, the SNR of the ultrasonic waveform signal has been significantly improved. This reduces the area of high-energy beamforming and mitigates the impact of a large number of false peaks on imaging positioning. The denoised signal is imaged and positioned using the improved WFB algorithm, and the results are shown in Figure 32.

Figure 32 shows the three-dimensional positioning results of the transformer patrol robot. The positioning results reveal that there is a maximum energy area at Z = 726 mm. We sliced the three-dimensional space positioning map of the sound source at Z = 726 mm. The slice map shows a minimal area of high energy. The coordinates of the maximum energy point are (234 mm, 52 mm), which is basically the actual location of the sound source.

In order to further verify the positioning accuracy of the three-dimensional positioning method at different positions of the patrol robot sound source, 14 groups of ultrasonic signals were collected at position 1 and position 2, respectively. Then, the improved WFB algorithm was used to localize the above signals, and the results are shown in Table 2 and Table 3.

The positioning results show that the relative error of the three-dimensional spatial positioning of the transformer patrol robot is less than 3.47%, and the maximum positioning error is less than 2.6 cm.

Similarly, the MVDR algorithm and GCC-PHAT algorithm were used for localization at positions 1 and 2, respectively. The localization error analysis is shown in Figure 33 and Figure 34.

As shown in Figure 33 and Figure 34, under different sound source positions, the localization performance of the WFB algorithm is superior to that of the MVDR algorithm and the GCC-PHAT algorithm in terms of both localization accuracy and localization value fluctuation, and it can meet the requirements of engineering applications. The localization performance of the method described in this paper has been verified.

## 5. Conclusions

In this paper, a three-dimensional spatial localization method for transformer patrol robots based on nine-element ultrasonic arrays, wavelet decomposition, and a WFB algorithm is proposed. To verify the performance of the method, the simulation results show that compared with the basic algorithm, the improved weighted filter-beamforming localization algorithm has stronger anti-noise and anti-reverberation capabilities, and improves localization accuracy in environments with strong noise and strong reverberation.

Based on simulation research, this paper built a three-dimensional positioning effect test platform for a transformer patrol robot and experimentally verified the effectiveness of the proposed positioning method. The localization results showed that the signal-to-noise ratio of the ultrasonic localization signal was significantly improved with semi-soft threshold wavelet denoising and reconstruction. This improved the ability to image small areas of high energy and reduced the impact of a large number of pseudo-peaks on imaging and localization. Using wavelet decomposition and the WFB algorithm resulted in a 3D spatial localization error of less than 3.47% and a maximum localization error of less than 2.6 cm, meeting engineering localization requirements.

## Figures and Tables

**Figure 1 sensors-25-05723-f001:**
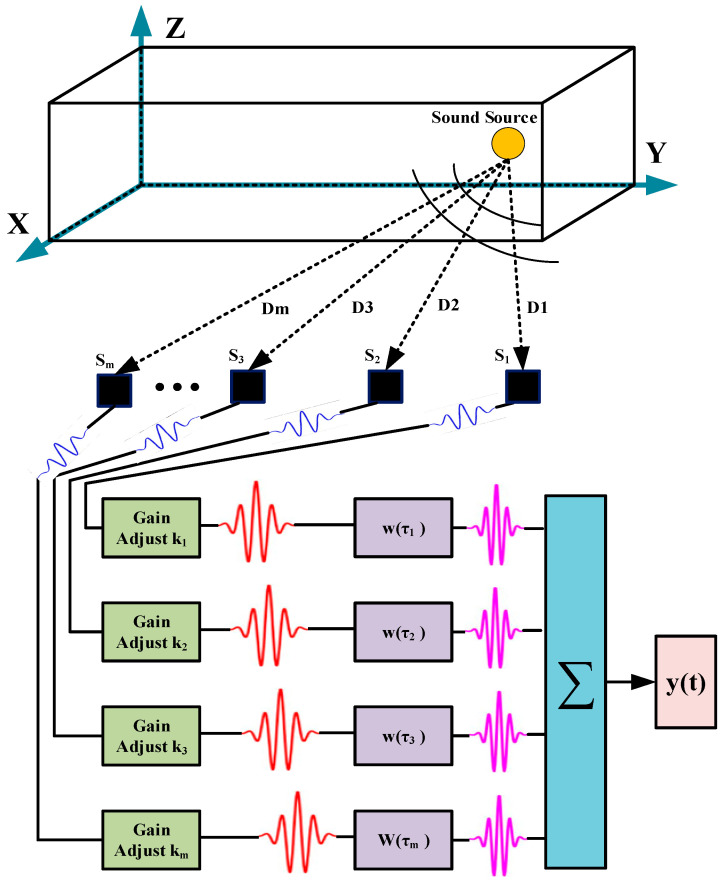
Architecture of the time delay-summing beamforming algorithm.

**Figure 2 sensors-25-05723-f002:**
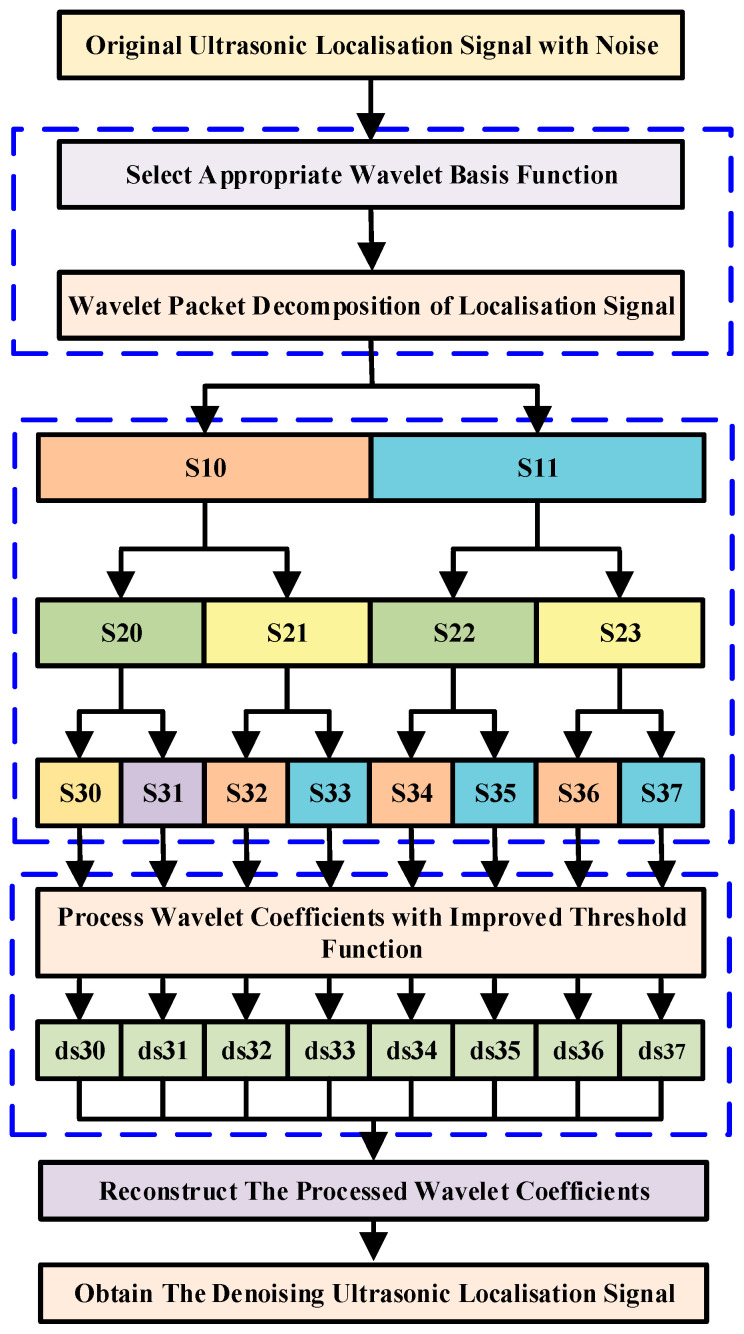
The denoising process of an ultrasonic localization signal.

**Figure 3 sensors-25-05723-f003:**
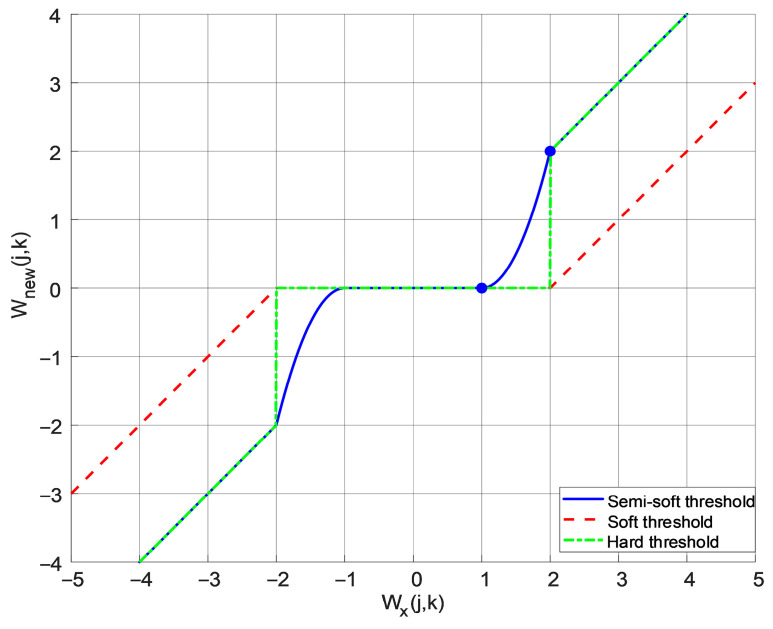
Threshold function curve ($\lambda_{1}=1$, $\lambda_{2}=2$).

**Figure 4 sensors-25-05723-f004:**
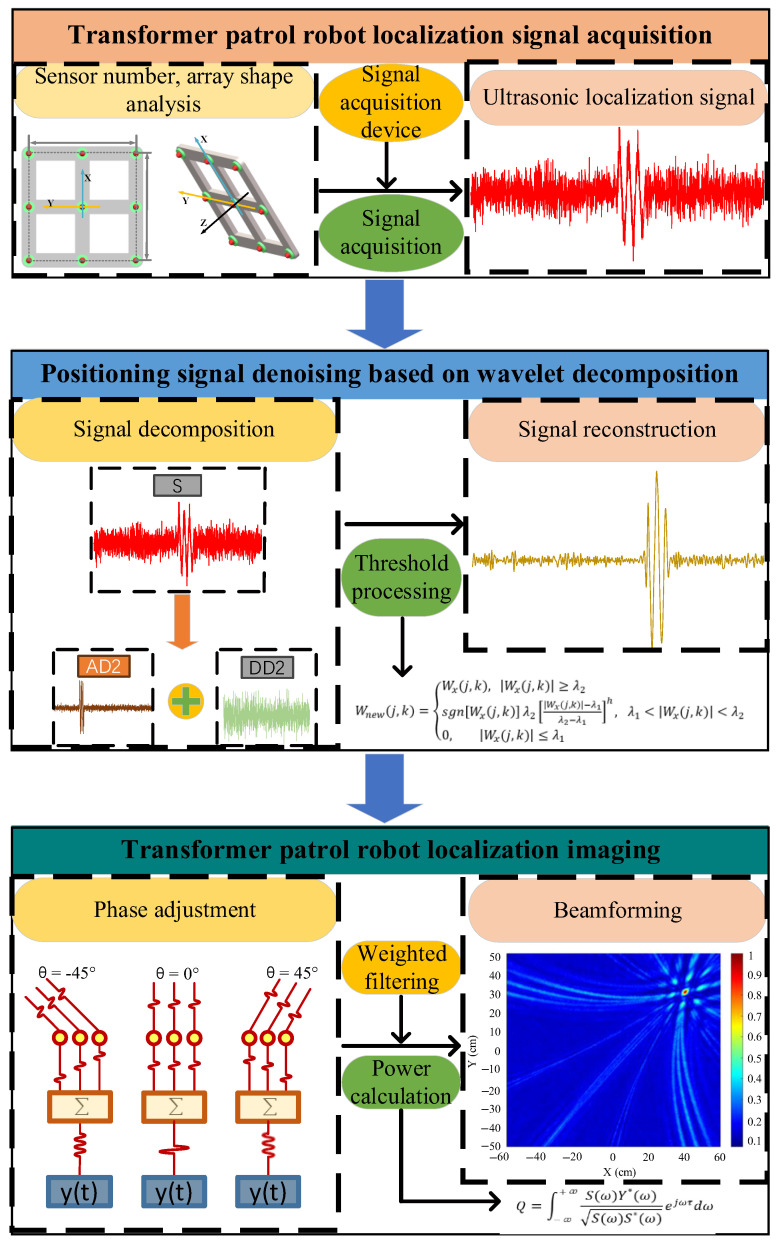
Three-dimensional spatial localization process of a transformer patrol robot.

**Figure 5 sensors-25-05723-f005:**
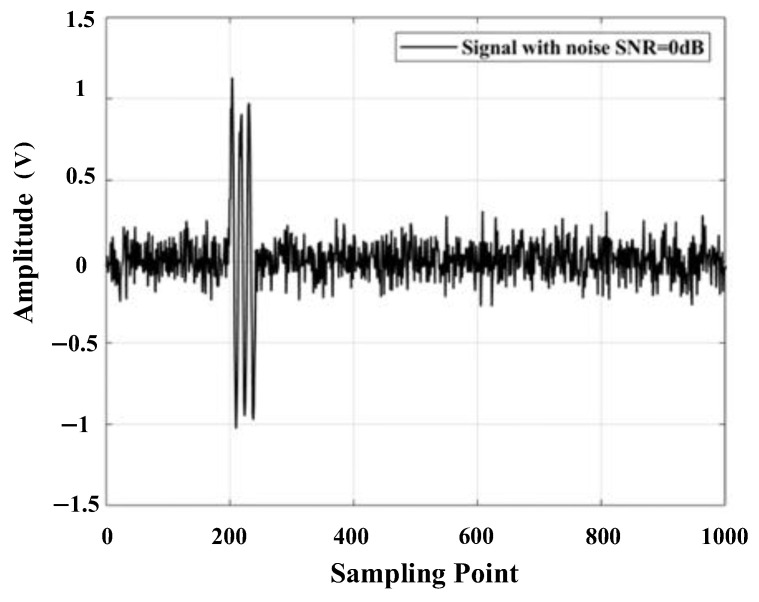
Simulated signal with an SNR of 0 dB.

**Figure 6 sensors-25-05723-f006:**
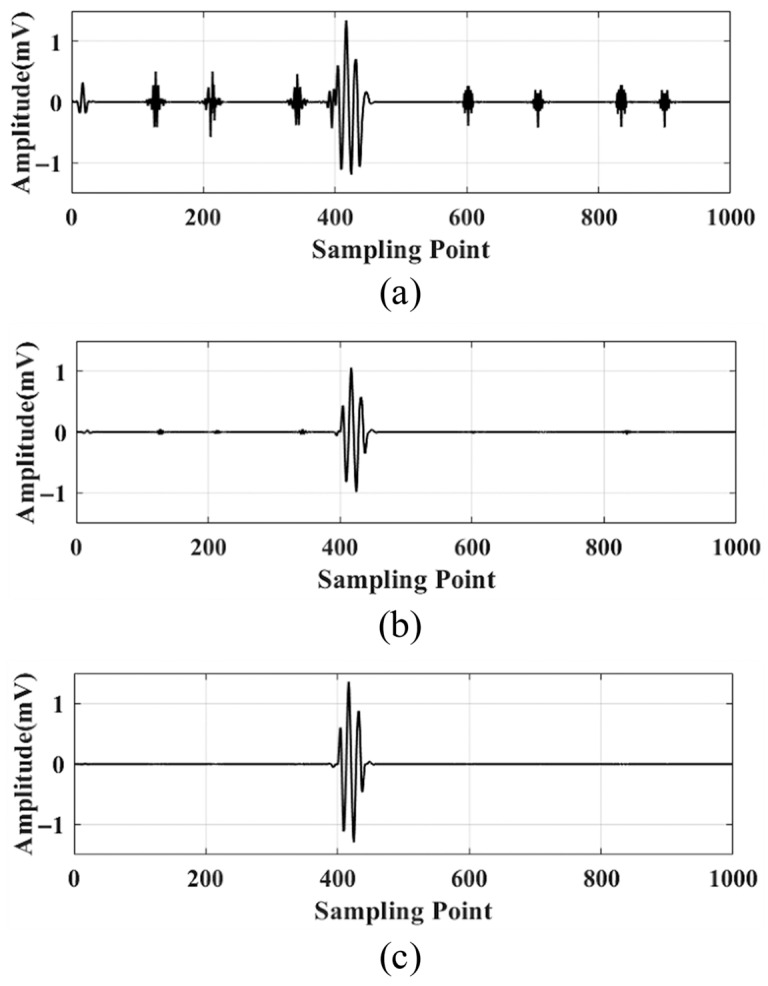
Comparison of the effect of different thresholding functions after denoising: (**a**) hard thresholding denoising; (**b**) soft thresholding denoising; (**c**) semi-soft thresholding denoising.

**Figure 7 sensors-25-05723-f007:**
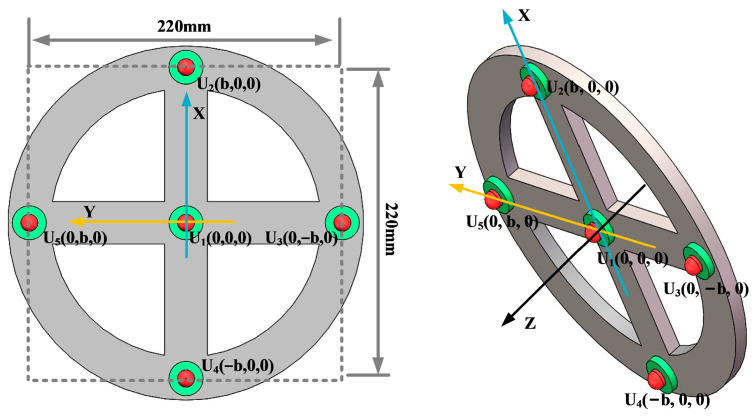
Circular ultrasonic array.

**Figure 8 sensors-25-05723-f008:**
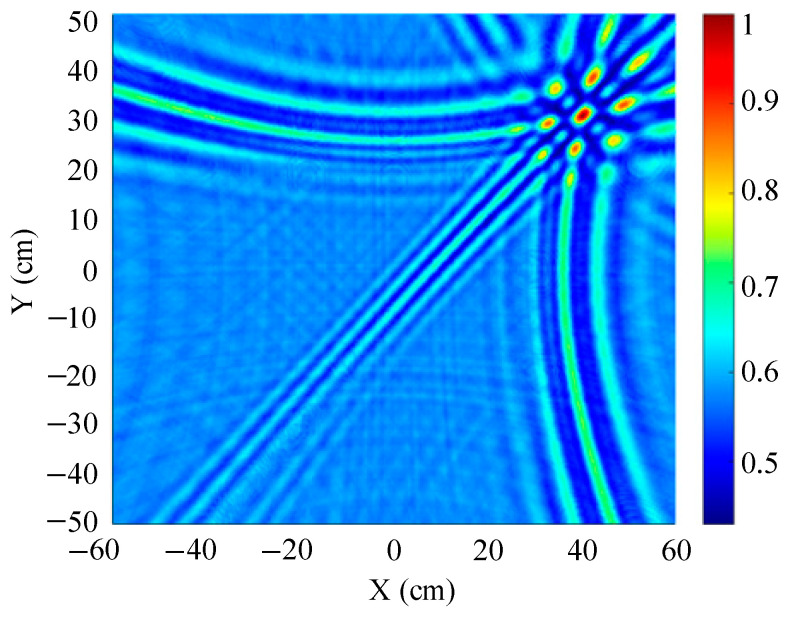
The localization result of the traditional beamforming algorithm.

**Figure 9 sensors-25-05723-f009:**
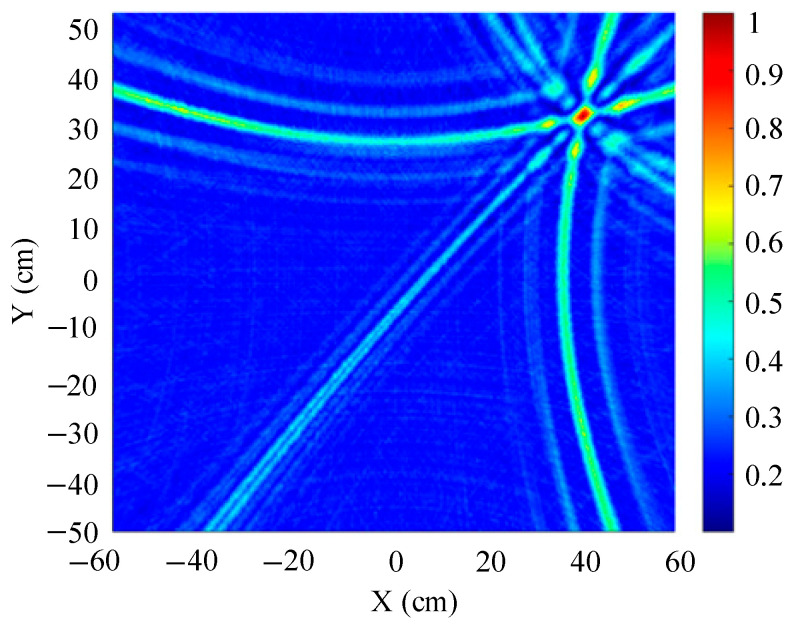
The localization result of the improved WFB algorithm.

**Figure 10 sensors-25-05723-f010:**
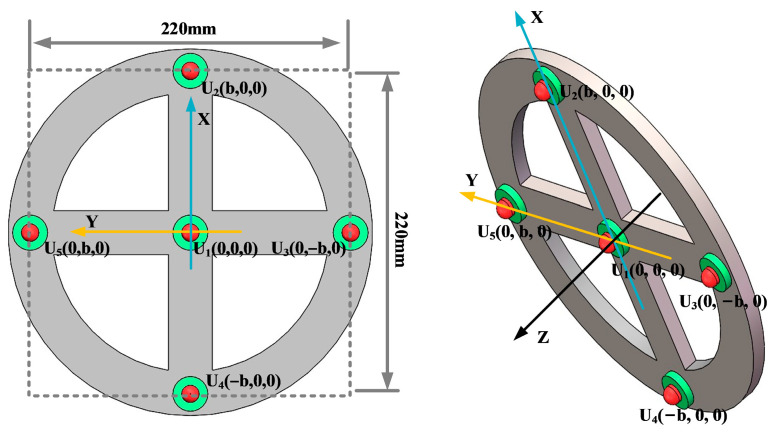
Circular ultrasonic array.

**Figure 11 sensors-25-05723-f011:**
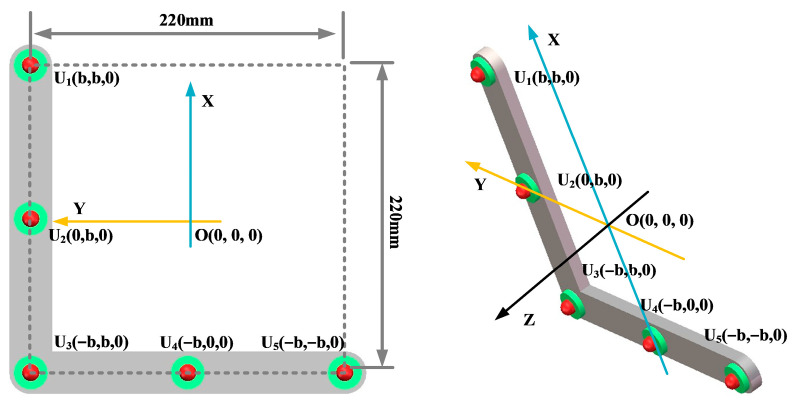
L-shaped ultrasonic array.

**Figure 12 sensors-25-05723-f012:**
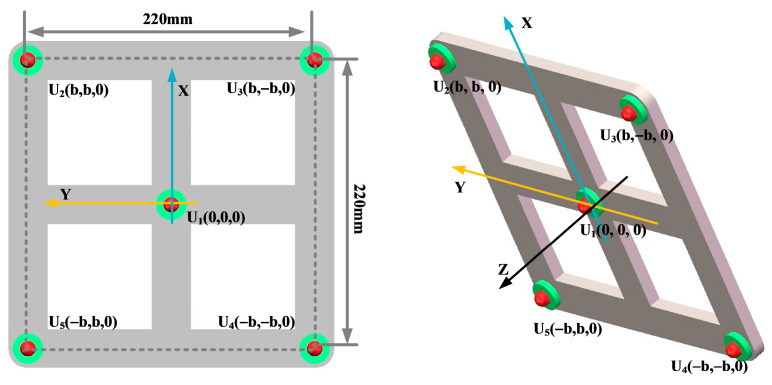
Square ultrasonic array.

**Figure 13 sensors-25-05723-f013:**
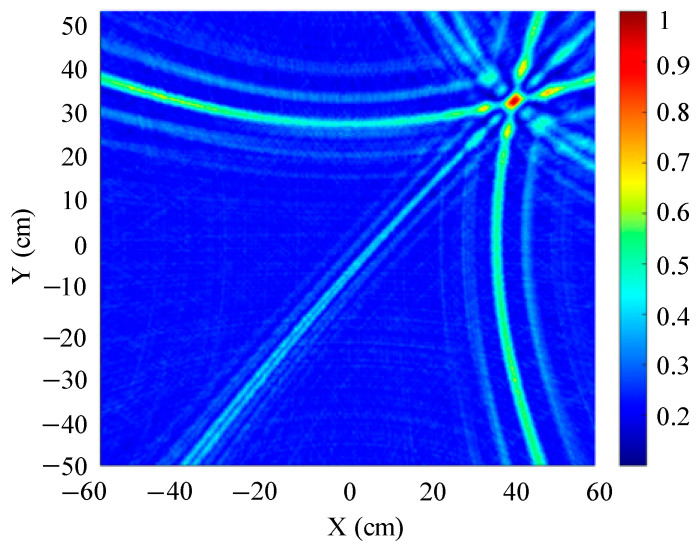
Positioning results of circular ultrasonic array.

**Figure 14 sensors-25-05723-f014:**
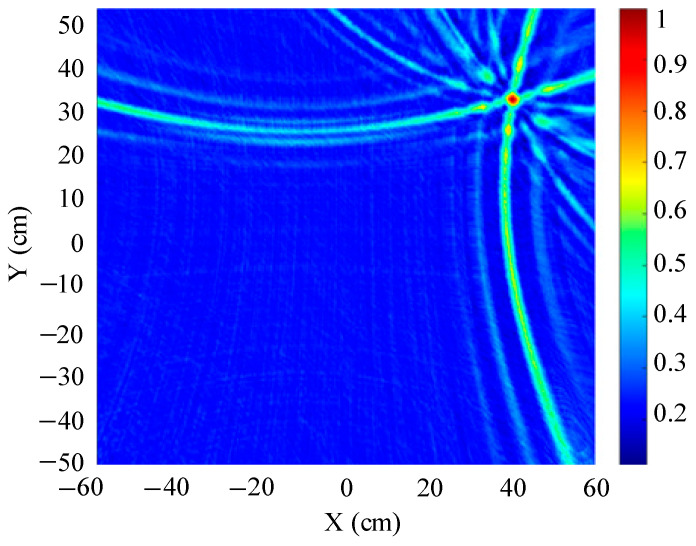
Positioning results of L-shaped ultrasonic array.

**Figure 15 sensors-25-05723-f015:**
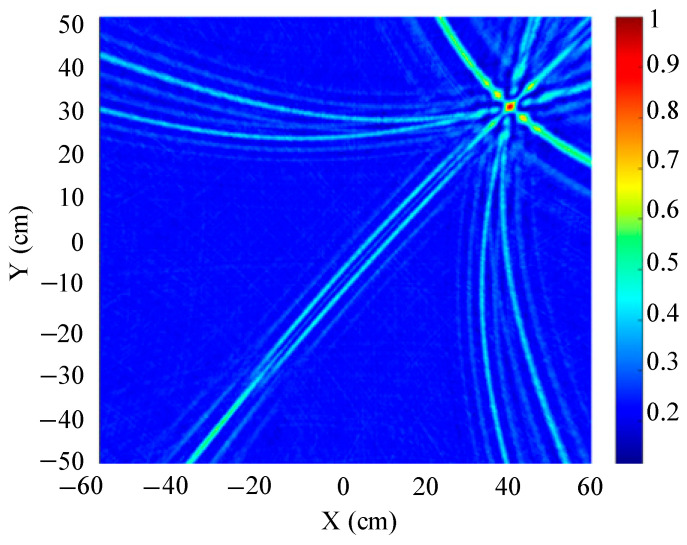
Positioning results of square ultrasonic array.

**Figure 16 sensors-25-05723-f016:**
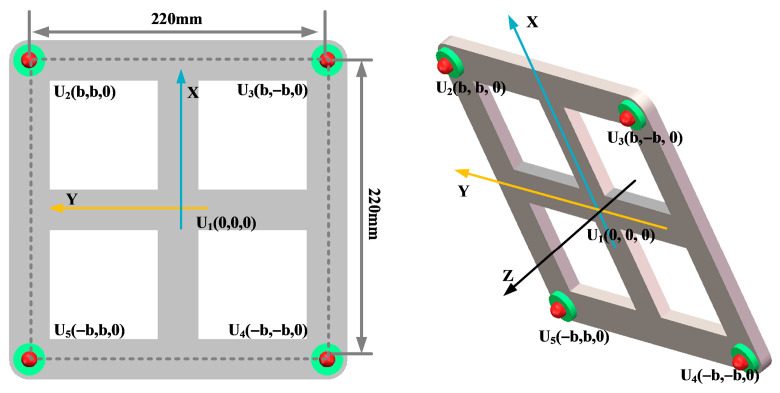
Four-element square arrays.

**Figure 17 sensors-25-05723-f017:**
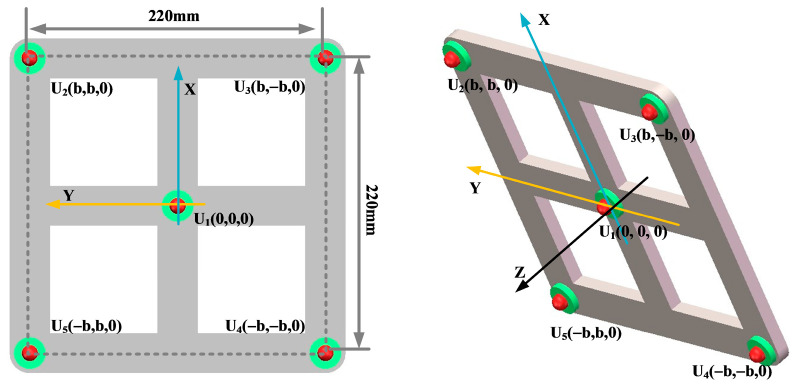
Five-element square arrays.

**Figure 18 sensors-25-05723-f018:**
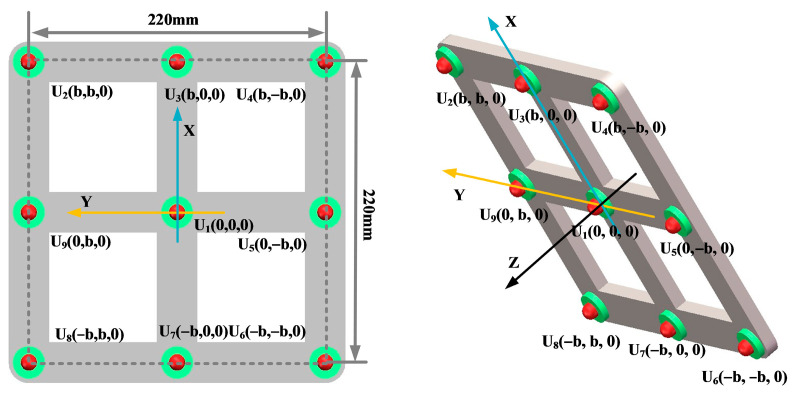
Nine-element square arrays.

**Figure 19 sensors-25-05723-f019:**
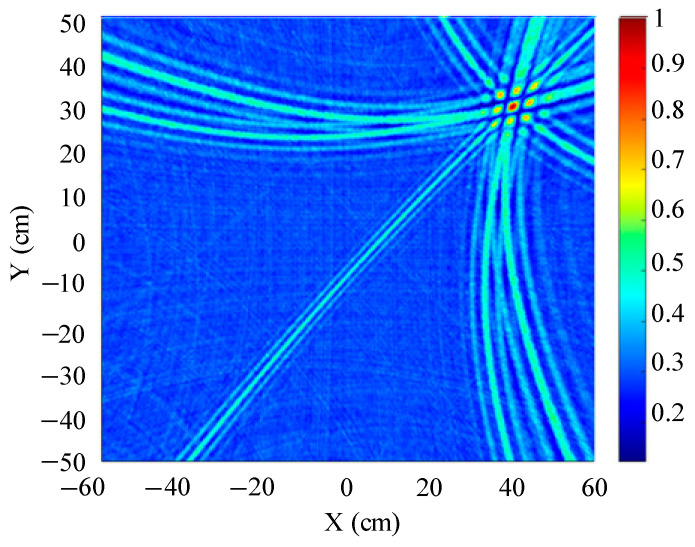
Positioning results of four-element square arrays.

**Figure 20 sensors-25-05723-f020:**
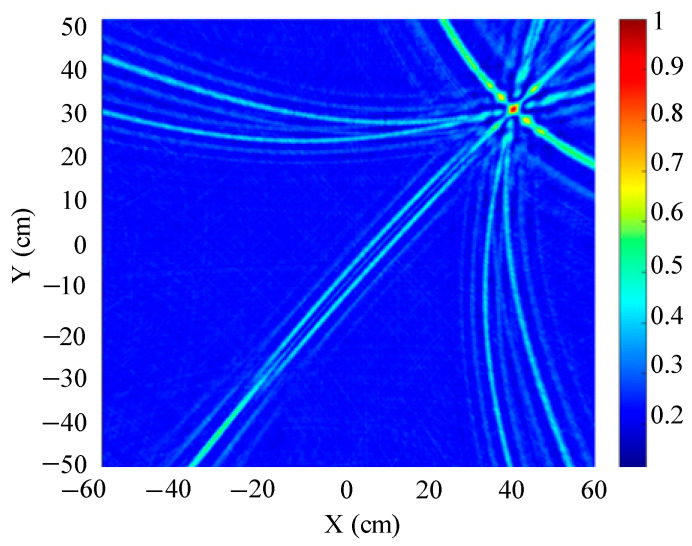
Positioning results of five-element square arrays.

**Figure 21 sensors-25-05723-f021:**
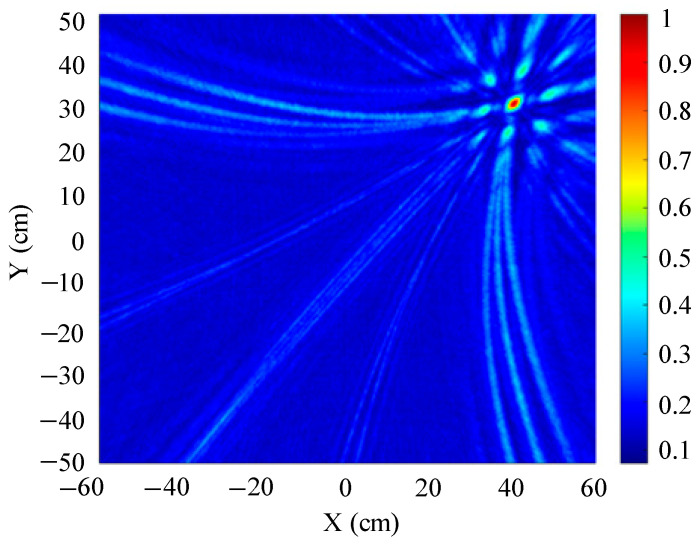
Positioning results of nine-element square arrays.

**Figure 22 sensors-25-05723-f022:**
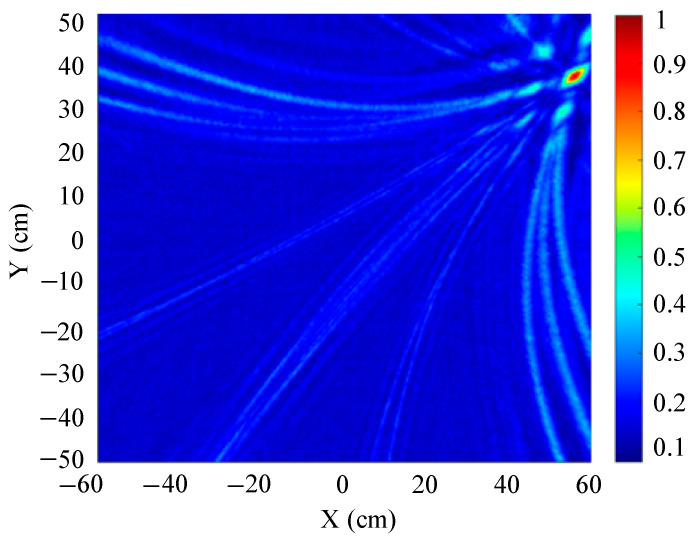
Sound source position is set as (560 mm, 360 mm, 600 mm).

**Figure 23 sensors-25-05723-f023:**
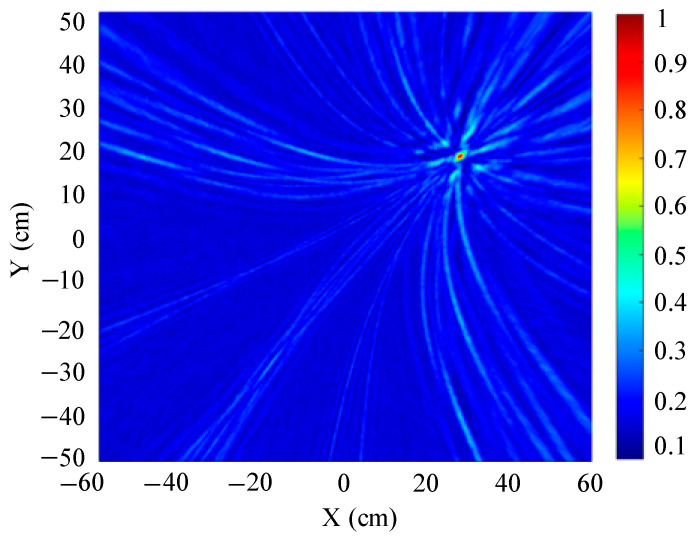
Sound source position is (280 mm, 180 mm, 300 mm).

**Figure 24 sensors-25-05723-f024:**
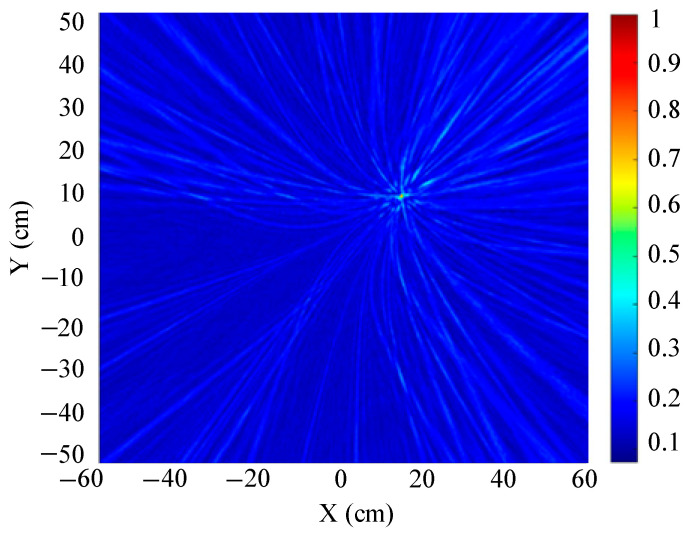
Sound source position is (140 mm, 90 mm, 150 mm).

**Figure 25 sensors-25-05723-f025:**
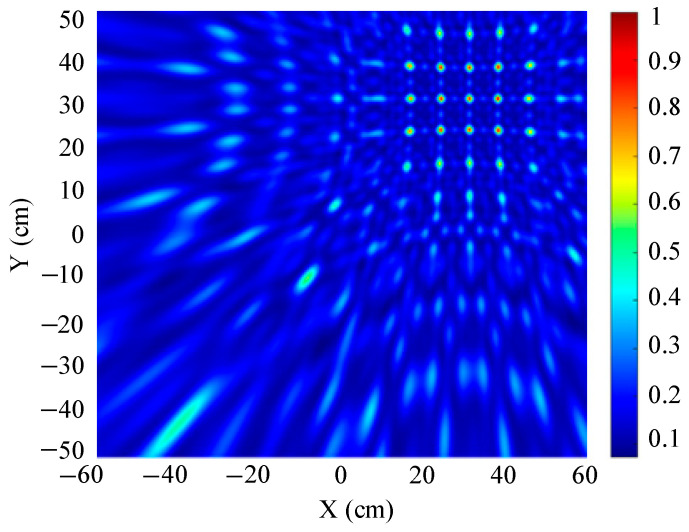
MVDR algorithm localization results when SNR = 0 dB.

**Figure 26 sensors-25-05723-f026:**
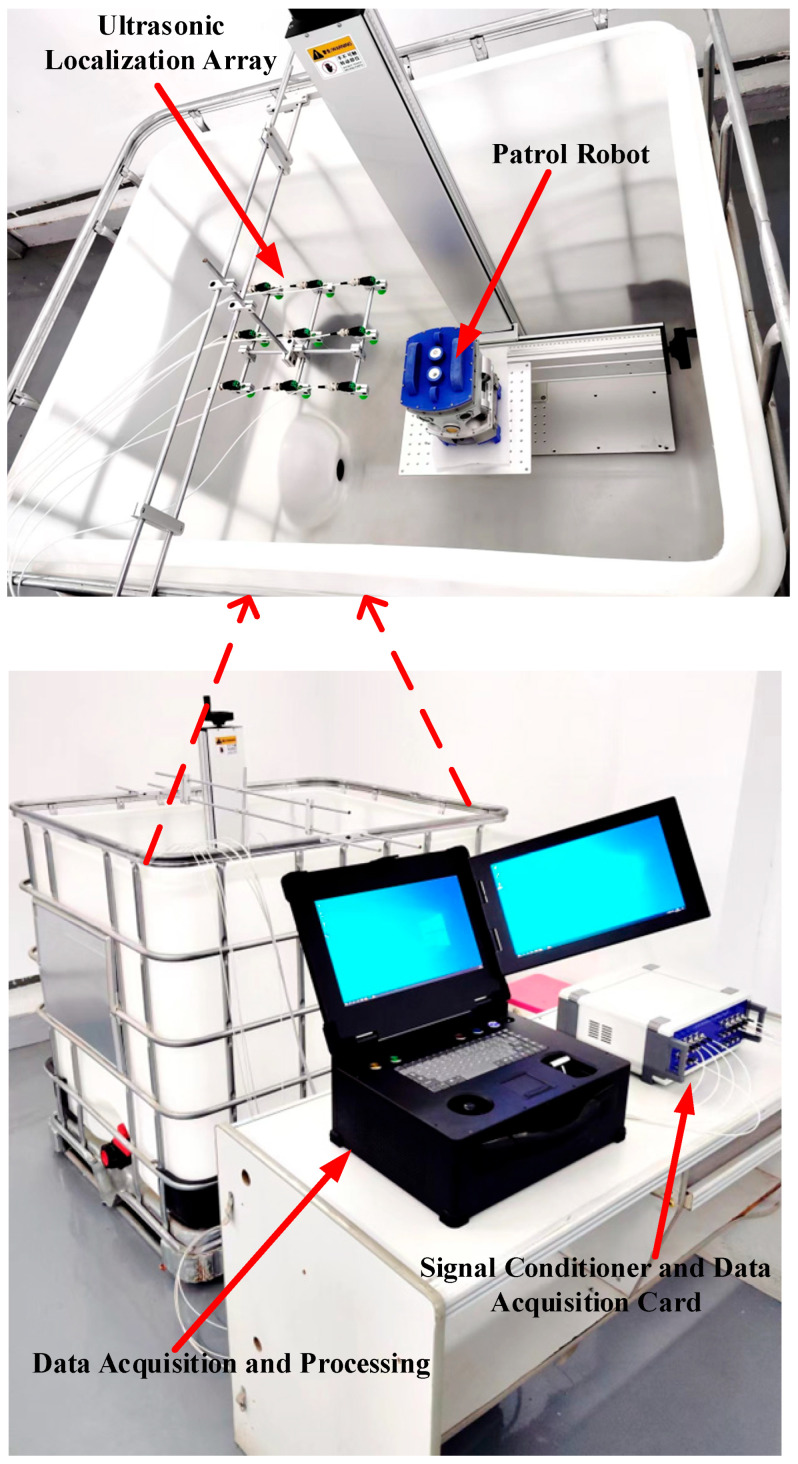
Three-dimensional positioning test platform for a patrol robot.

**Figure 27 sensors-25-05723-f027:**
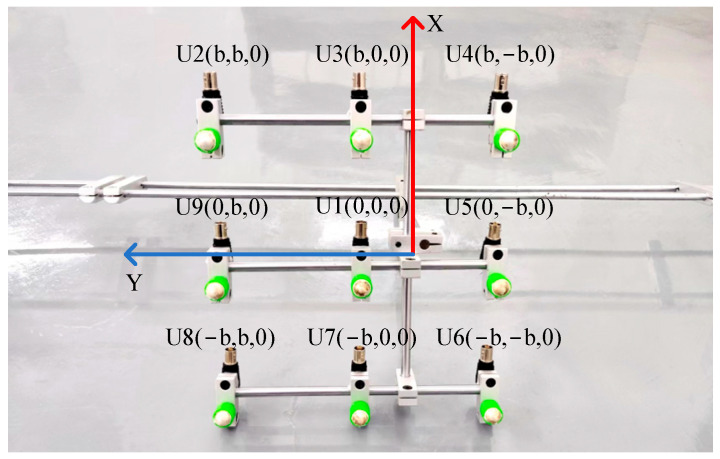
Nine-array rectangular ultrasonic sensor array.

**Figure 28 sensors-25-05723-f028:**
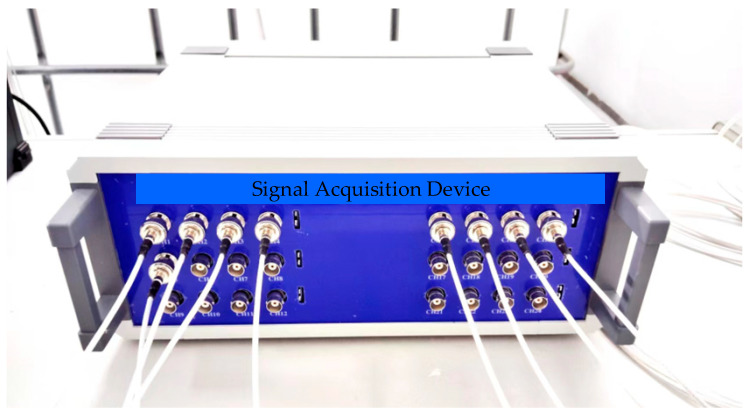
Signal acquisition device.

**Figure 29 sensors-25-05723-f029:**
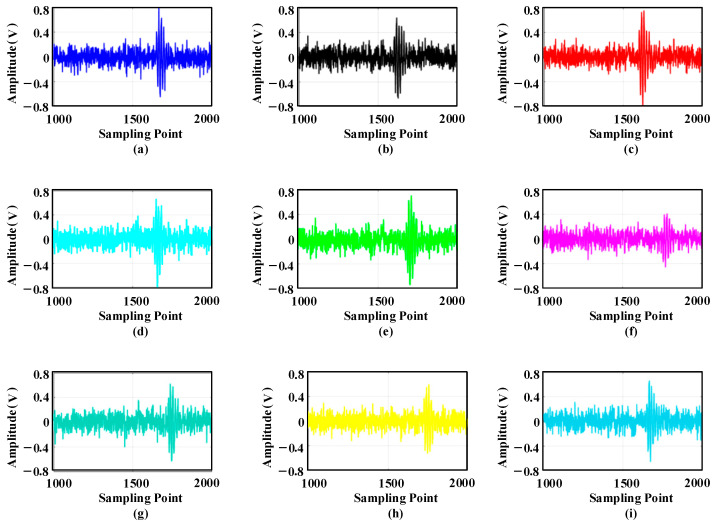
Ultrasonic signal waveform acquired by sensor device: (**a**) sensor 1; (**b**) sensor 2; (**c**) sensor 3; (**d**) sensor 4; (**e**) sensor 5; (**f**) sensor 6; (**g**) sensor 7; (**h**) sensor 8; (**i**) sensor 9.

**Figure 30 sensors-25-05723-f030:**
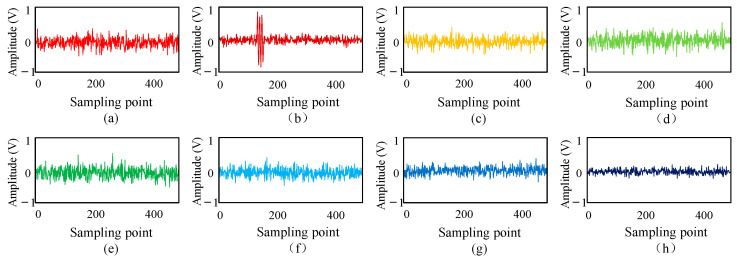
Wavelet packet decomposition coefficients of ultrasonic signals: (**a**) a30; (**b**) a31; (**c**) a32; (**d**) a33; (**e**) a34; (**f**) a35; (**g**) a36; (**h**) a37.

**Figure 31 sensors-25-05723-f031:**
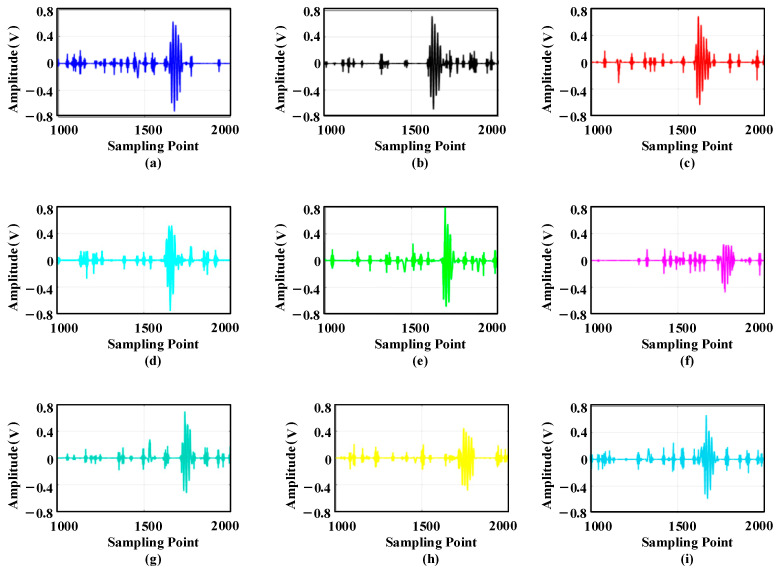
Waveform reconstruction by wavelet packet denoising: (**a**) sensor 1; (**b**) sensor 2; (**c**) sensor 3; (**d**) sensor 4; (**e**) sensor 5; (**f**) sensor 6; (**g**) sensor 7; (**h**) sensor 8; (**i**) sensor 9.

**Figure 32 sensors-25-05723-f032:**
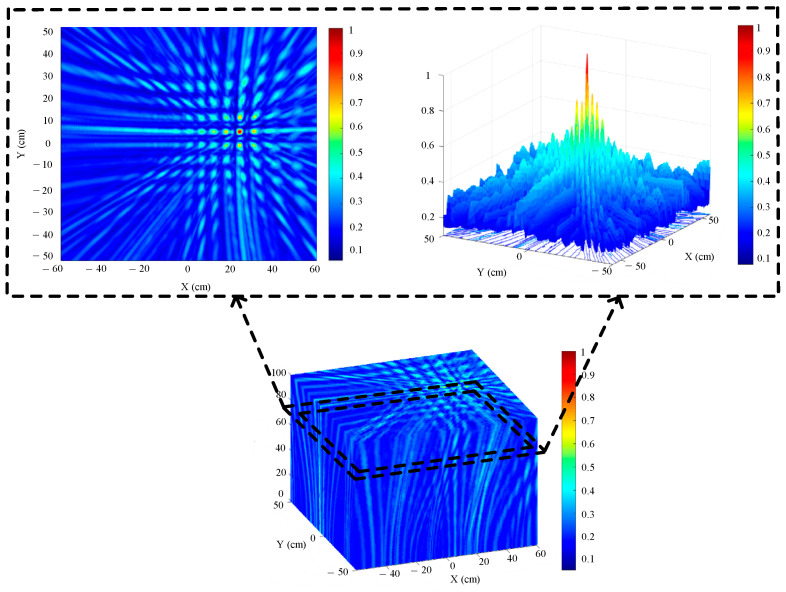
Three-dimensional positioning results of the patrol robot.

**Figure 33 sensors-25-05723-f033:**
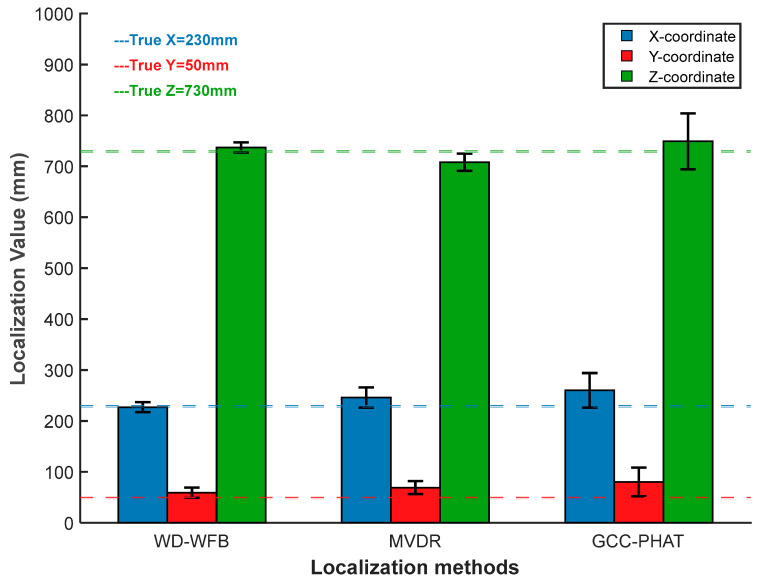
Localization error analysis at position 1.

**Figure 34 sensors-25-05723-f034:**
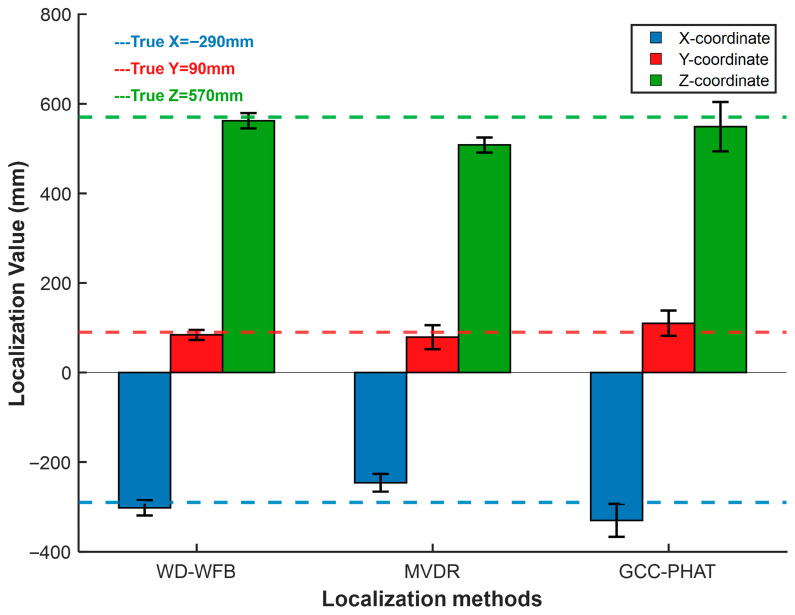
Localization error analysis at position 2.

**Table 1 sensors-25-05723-t001:** Comparison for evaluating the denoising effect of signals with an SNR of 0 dB.

Threshold Function	SNR/dB	RMSE	NCC
Original signal with noise	0	0.315	0.562
Denoising with hard threshold function	6.982	0.114	0.743
Denoising with soft threshold function	9.717	0.092	0.803
Denoising with semi-soft threshold function	11.206	0.047	0.928

**Table 2 sensors-25-05723-t002:** Localization results at position 1.

Position 1 (mm)	Localization Results (mm)	Absolute Localization Error (mm)	Relative Localization Error
(230, 50, 730)	(238, 54, 746)	(8, 4, 16)	2.39%
(230, 50, 730)	(242, 58, 740)	(12, 8, 10)	2.29%
(230, 50, 730)	(236, 54, 736)	(6, 4, 6)	1.22%
(230, 50, 730)	(234, 52, 720)	(4, 2, 10)	1.43%
(230, 50, 730)	(228, 52, 710)	(2, 2, 20)	2.63%
(230, 50, 730)	(244, 46, 738)	(14, 4, 8)	2.17%
(230, 50, 730)	(218, 58, 712)	(2, 8, 18)	3.01%
(230, 50, 730)	(234, 54, 704)	(4, 4, 26)	3.47%
(230, 50, 730)	(240, 42, 726)	(10, 8, 4)	1.75%
(230, 50, 730)	(244, 54, 732)	(14, 4, 2)	1.92%
(230, 50, 730)	(234, 52, 726)	(4, 2, 4)	0.78%
(230, 50, 730)	(238, 54, 730)	(8, 4, 0)	1.17%
(230, 50, 730)	(218, 56, 742)	(12, 6, 12)	2.35%
(230, 50, 730)	(228, 50, 704)	(2, 0, 26)	3.40%

**Table 3 sensors-25-05723-t003:** Localization results at position 2.

Position 2 (mm)	Localization Results (mm)	Absolute Localization Error (mm)	Relative Localization Error
(−290, 90, 570)	(−290, 92, 570)	(0, 2, 0)	0.31%
(−290, 90, 570)	(−294, 94, 578)	(4, 4, 8)	1.52%
(−290, 90, 570)	(−294, 96, 578)	(4, 6, 8)	1.67%
(−290, 90, 570)	(−296, 96, 584)	(6, 6, 4)	1.45%
(−290, 90, 570)	(−290, 96, 578)	(0, 6, 8)	1.55%
(−290, 90, 570)	(−296, 92, 584)	(6, 2, 14)	2.38%
(−290, 90, 570)	(−294, 94, 578)	(4, 4, 8)	1.52%
(−290, 90, 570)	(−290, 92, 570)	(0, 2, 0)	0.10%
(−290, 90, 570)	(−298, 94, 582)	(8, 4, 12)	2.32%
(−290, 90, 570)	(−296, 92, 584)	(6, 2, 14)	2.38%
(−290, 90, 570)	(−286, 88, 558)	(4, 2, 12)	1.98%
(−290, 90, 570)	(−298, 94, 574)	(8, 4, 4)	1.52%
(−290, 90, 570)	(−296, 96, 584)	(6, 6, 14)	2.53%
(−290, 90, 570)	(−294, 96, 584)	(4, 6, 14)	2.44%

## Data Availability

The data presented in this study are available on request from the corresponding author. The data are not publicly available due to privacy.

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
