# Peer review of "Ultrasonic Localization of Transformer Patrol Robot Based on Wavelet Transform and Narrowband Beamforming"

_sensors, 2025, doi:10.3390/s25185723_

Round 1
Reviewer 1 Report
Comments and Suggestions for Authors
The manuscript proposes a wavelet decomposition and weighted filter-beamforming (WD-WFB) algorithm for ultrasonic localization of patrol robots in transformer tanks. While the implementation appears technically sound and the simulation/experimental results demonstrate promising accuracy under specific test conditions, the work lacks a rigorous theoretical justification and comparative performance analysis against state-of-the-art ultrasonic localization methods. The reviewer’s major comments are as follows.
1.The improvements claimed for the WD-WFB algorithm over conventional methods are empirically shown but not theoretically analyzed. For instance, there is no derivation or formal discussion explaining why the proposed semi-soft threshold wavelet denoising significantly outperforms standard hard/soft thresholding in terms of preserving phase accuracy, SNR enhancement, or resolution gain.
2.Similarly, while Figures 5–6 and 16–18 show improved localization under certain conditions, the paper does not quantitatively compare the proposed method against other high-resolution algorithms such as MUSIC, MVDR, or GCC-PHAT under matched test conditions (e.g., varying SNR, multipath, or reverberant settings).
3.The parameter sensitivity (e.g., threshold values λ1, λ2, array geometry specifics) and generalizability of the proposed method under real transformer operational conditions are not discussed. This limits the engineering relevance and repeatability of the proposed approach.
Reviewer 2 Report
Comments and Suggestions for Authors
This paper proposes a method of localization of navigating robot based on beamforming in consideration of physical environment of a power transformer. However, this paper does not seem to be ready for being reviewed in the sense that its editing and readability are too poor to be reviewed.
- Authors need to rewrite abstract such that it can deliver its research clearly. For example, Then --> Finally --> Then which are quite awkward structure in abstract.
- Please justify why the inside of the transformer has high-noise environment within transformers when the medium is ultrasonic wave and it is shielded by design.
- Please add description on gradp and divu in equation (4)
- line 137 has typo.
- Editing needs to be improved significantly. In addition to strange position of some symbols in a line, authors seem to reuse the same symbol for different purpose even without mentioning what it is. For example, Is psai in (17) same as psai (19)?
- (17) can not be power. Authors has critical error in important equation. In addition, the 2nd inequality can not be hold since Ψ(ω) is 1/|S(w)] by definition while it is defined as the weighted filter function.
It is readable. However, some awkward expression give vagueness.
Reviewer 3 Report
Comments and Suggestions for Authors
The article is devoted to a technically very interesting topic - localization of a miniature robot moving inside a transformer. The authors propose to use ultrasound for this, which is reasonable and technically justified.
The article begins with a brief introduction to the laws of ultrasound propagation in the environment. The authors provide analytical expressions necessary to explain the key idea of the work - the use of an array of ultrasonic sensors to determine the direction of the sound source (robot).
The article describes all stages of the research in detail. Several configurations of microphone arrays are described. The article ends with experimental measurements that give a good result. The authors claim virtually centimeter-level positioning accuracy, which is very good.
The drawings are of good quality and are clearly visible both electronically and in printed form.
In essence, I have no comments on the article.
The only comments are related to the layout of the text:
1. The titles of Sections 2 and 2.2 in the current version are torn off from the main text.
2. The last row of table two is torn off from the main body of the table.
3. There are large gaps on some pages: on pages 6, 9, 18, 19, 20
4. Perhaps it is worth considering reformatting (more compact presentation) the figures where the receiver arrays are shown: they take up a lot of space, there are many of them, but in fact they do not transmit very much information. This resembles a very detailed technical report.
5. The same applies to the figures: positioning results of ultrasonic arrays. Are they all really needed, if so, then maybe they can be combined somehow?
Reviewer 4 Report
Comments and Suggestions for Authors
The authors propose a method that integrates wavelet packet decomposition and weighted filter beamforming (WD-WFB) to enhance ultrasonic localization in high-noise, reverberant environments, such as those found in transformer tanks.
However, before publication some modifications must be made.
- First, although the semi-soft thresholding function is presented as innovative, the paper lacks a detailed theoretical justification or statistical performance comparison against soft and hard thresholding techniques (e.g., RMSE, SNR, or MAE). This omission limits readers' ability to assess the benefits of the proposed function. Additionally, although the wavelet and beamforming techniques are integrated effectively, the signal processing pipeline is not clearly visualized or explained. Readers would benefit from a block diagram or a more explicit, step-by-step explanation of the process from signal acquisition to position estimation.
-
The details of the hardware and system implementation are also vague. The specifications of the ultrasonic sensors, such as frequency range, bandwidth, beam width, and sensitivity, are not included. Similarly, the description of the robot's capabilities, such as processing hardware, localization rate, and on-board power constraints, is insufficient to assess real-time feasibility. Additionally, the experimental setup assumes a static, clean oil environment, but does not address variations in temperature, oil density, or dynamic obstacles, all of which can affect ultrasonic propagation and reflectivity. The scalability and robustness of the method in diverse or scaled-up transformer tanks are not discussed, nor is its performance evaluated with moving robots or changing conditions.
-
The paper's presentation suffers from inconsistencies in terminology, notation, and formatting. For example, terms such as "weighted filter beamforming" and "weighted filtering beamforming" are used interchangeably. Some figures lack axis labels and clear legends, which reduces their clarity. The manuscript contains numerous grammatical errors and awkward phrasing, especially in technical descriptions, which would benefit from thorough proofreading by a native English speaker. Equation explanations are often brief, and several symbols are either undefined or insufficiently contextualized.
Therefore, I recommend major revisions before the work can be considered for publication.
Round 2
Reviewer 1 Report
Comments and Suggestions for Authors
No comments.
Author Response
Thank you for your work about our manuscript.
Reviewer 2 Report
Comments and Suggestions for Authors
This manuscript has been improved significantly through revision. However, there are still some issues and overlooked and inaccurate expressions. More importantly, the contribution of the proposed preprocessing seems to be limited in the sense that theoretical or empirical superiority to the existing preprocessing has not been well presented. Authors made comparison with existing beamforming methods. However, the main contribution can be preprocessing. However, the proposed preprocessing approaches are quite heuristic.
1.Abstract still needs improvement. What can be know from knowing SNR, NCC and RMSE? It will be better to add how much improvement has been made in reference to the base line method. "The denoising effect was significantly improved compared to the traditional threshold function" Author also needs to provide quantitative evidence from the experiment to support this argument.
2. The authors argued in the response "In complex on-site environments, sensor arrays detect ultrasonic localization signals that are easily affected by various types of noise interference. These noises may originate from the operation of surrounding equipment, environmental electromagnetic fields, or other random noise signals" How is this physically possible? How can electromganetic fields works as noise to ultrasonic wave? Response to the comment 2 does not appeart to be scientifically sound.
3. This is not correct "This means that in the frequency domain, each frequency component of the output signal is weighted according to the amplitude of the reference signal S(ω). Components with smaller frequency amplitudes are amplified, while components with larger frequency amplitudes are suppressed." S(w)/sqrt(S(w)S(w)*) works as phase matching filter to output Y(w).
4. The statements sounds to be confusing and inaccurate "Search all possible spatial locations; the location where Q reaches its maximum value is the sound source location. Compared to the basic algorithm, the improved localization algorithm based on WFB highlights true peaks, has stronger noise and reverberation resistance, and improves the localization accuracy of the algorithm in environments with strong noise and strong reverberation."
5. Authors seems to ignore fundamental limitation of the ultrasonic array. To the best of author's knowledge, The maximum field of view of ultrasonic sensor is liekly to be 180 degree. However authors argue that the proposed method supports localization in 3D space. While it may be the case with array for electromagnetic waves, it may not be the case with the ultrasonic waves. To the best of the reviewer's projection, the proposed methods can localize the object in one direction while it can not in the opposite direction. The considered array can be applicable at the bottom or at the top of the structure which can reduce the required field of view by half physically.
6. The reviewer believes that the contribution of this research is lying on the preprocessing rather than beamforming algorithm itself. While authors made comparison with existing beamforming algorithms, it is doubtful that proper preprocessing was considered for fair comparison.
Less...
Reviewer 4 Report
Comments and Suggestions for Authors
Compared to the original submission, the revised version shows significant improvements in clarity, organization, and presentation. The processing pipeline is more clearly explained, the figures are cleaner, and the references are handled more consistently. However, key scientific gaps remain, including a lack of statistical rigor, insufficient validation of the semi-soft thresholding function, and limited discussion of robustness and deployment scalability. These issues limit the manuscript's strength and impact.
The paper still lacks statistical rigor in its evaluation: the reported localization errors are not supported by error bars, standard deviations, or confidence intervals, which makes it difficult to assess the robustness of the claims. The newly proposed semi-soft thresholding function is described but not quantitatively compared against standard hard or soft thresholding approaches; this weakens the argument for its novelty and superiority. The experiments are limited to a narrow test environment, without exploring how the method performs under different transformer conditions, noise levels, or scaling scenarios. Hardware and system details remain insufficient, with vague descriptions of sensor specifications, calibration procedures, and robot integration, which hinders reproducibility. Figures, although improved, still lack complete axis labels, units, and fully self-contained captions, reducing their clarity. Finally, while the literature review is somewhat expanded, it still does not adequately compare the proposed approach against the latest ultrasonic localization and beamforming methods, nor does it explicitly highlight the contribution’s uniqueness relative to existing work.
The authors should add quantitative statistical validation, expand robustness testing, clarify sensor and hardware details, and improve figure quality and language. These changes would make the work suitable for publication.
